# Circular RNAs in Ischemic Stroke: Biological Role and Experimental Models

**DOI:** 10.3390/biom13020214

**Published:** 2023-01-22

**Authors:** Chiara Siracusa, Jolanda Sabatino, Isabella Leo, Ceren Eyileten, Marek Postuła, Salvatore De Rosa

**Affiliations:** 1Department of Medical and Surgical Sciences, Magna Graecia University, 88100 Catanzaro, Italy; 2Department of Children and Woman’s Health, University of Padua, 35121 Padua, Italy; 3Department of Cardiology, Royal Brompton and Harefield Hospitals, Guy’s and St Thomas’ NHS Foundation Trust, London SW3 5NP, UK; 4Center for Preclinical Research and Technology CEPT, Department of Experimental and Clinical Pharmacology, Medical University of Warsaw, 02-097 Warsaw, Poland; 5Genomics Core Facility, Center of New Technologies, University of Warsaw, 02-097 Warsaw, Poland

**Keywords:** stroke, circular RNAs, non-coding RNAs, ischemia

## Abstract

Ischemic stroke is among the leading causes of morbidity, disability, and mortality worldwide. Despite the recent progress in the management of acute ischemic stroke, timely intervention still represents a challenge. Hence, strategies to counteract ischemic brain injury during and around the acute event are still lacking, also due to the limited knowledge of the underlying mechanisms. Despite the increasing understanding of the complex pathophysiology underlying ischemic brain injury, some relevant pieces of information are still required, particularly regarding the fine modulation of biological processes. In this context, there is emerging evidence that the modulation of circular RNAs, a class of highly conserved non-coding RNA with a closed-loop structure, are involved in pathophysiological processes behind ischemic stroke, unveiling a number of potential therapeutic targets and possible clinical biomarkers. This paper aims to provide a comprehensive overview of experimental studies on the role of circular RNAs in ischemic stroke.

## 1. Introduction

Ischemic stroke (IS) is the leading cause of death and disability worldwide. IS is characterized by occlusion of the cerebral artery with consequent death and functional loss of brain cells [1]. Damage to brain cells can manifest through widespread symptoms, including vision problems, sudden weakness, and speech disturbances accompanied by muscle paralysis. The most common causes of IS are atherothrombotic occlusion of the large arteries, cerebral embolism, and non-thrombotic occlusion of small deep cerebral arteries.

The subsequent cellular injury is then caused by a number of complex pathophysiological events, including excitotoxicity, oxidative stress, inflammation, and apoptosis [2]. Additionally, stroke patients have an increased risk of neurodegenerative diseases such as Alzheimer’s disease and multiple sclerosis. However, despite some insights into the pathophysiology being available in the literature, the fine mechanisms underlying neuronal death caused by cerebral ischemia and their modulation have been not fully understood [3].

Further studies are needed to understand the cellular and molecular mechanisms involved in stroke injuries in order to find new biomarkers and potential therapeutic targets.

Numerous studies have shown how the expression and modulation of specific circularRNAs (circRNAs) are associated with many pathological and physiological processes. In particular, they seem to play a fundamental role in the development and progression of specific neurological diseases, including IS [4].

The circRNAs are a class of non-coding RNA with a highly stable and conserved closed-loop structure. CircRNAs have multiple functions and their presence in a stable form has been already established in different biological fluids, particularly in peripheral blood using exosomes as vectors. Interestingly, it has been proven that exosomes are able to cross the blood-brain barrier (BBB).

The presence of circRNA can be detected through blood samples, thus allowing the diagnosis of diseases through non-invasive tests. Most circRNAs can act as microRNA (miRNA) sponges to regulate endogenous gene expressions across the competing RNA network. A large number of studies have shown that microRNAs have a significant role in IS [5,6,7]. In particular, thanks to the interaction with miRNAs, circRNAs are key in gene expression regulation by modulating specific pathways, with consequences on the progression of some diseases [8,9,10].

In this review, we aim to provide an overview of the available studies that evaluate and analyze the role of circRNAs in IS as potential new diagnostic and prognostic biomarkers.

## 2. Article Search and Selection Results

This review was performed in accordance with PRISMA (Preferred Reporting Items for Systematic Reviews and Meta-Analyses) guidelines. We performed a literature search of the PubMed and Scopus databases using the terms “rna, circular” [MeSH Terms] OR (“rna” [All Fields] AND “circular” [All Fields]) OR “circular rna” [All Fields] OR “circrna” [All fields] OR “circrnas” [All fields]) AND (“stroke” [MeSH terms] OR “stroke” [All fields] OR “strokes” [All fields] OR “stroke s” [All fields]. The research included all in vivo and in vitro experimental studies in order to identify the circRNAs most involved in inflammation, angiogenesis, and ischemic stroke. The articles used in this work were independently selected according to the PRISMA protocol by two researchers. The protocol for this systematic review has been registered on the https://aspredicted.org/ repository of Pennsylvania University accessed on 12 January 2022 (registration number: 118582).

Altogether, 45 studies fulfilling the selection criteria were finally included in our systematic review. The article selection process, from search results to selection, is reported in detail in Figure 1, while a complete list and a synthetic description of all studies included is reported in Table 1.

## 3. Middle Cerebral Arterial Occlusion

Middle Cerebral Arterial Occlusion (MCAO) is one of the most common experimental models for in vivo stroke research. The MCAO model involves the occlusion of the common carotid artery by the introduction of a surgical filament into the artery with an interruption of blood flow and consequent cerebral infarction. The model can be used to reproduce a permanent or transient focal cerebral IS, depending on the duration and location of the occlusion. The suture can be removed after 30 up to 120 min, to obtain an ischemia/reperfusion damage model. Monitoring of the cortical blood flow by means of laser doppler or ultrasound-based techniques is helpful to verify the correct execution of the procedure and to record the onset of cerebral ischemia.

Using this model, high-throughput sequencing revealed differential expression of 37548 circRNAs in the ischemic area as compared to the ipsilateral healthy thalamus at 14 days. Mmu_circ: chr2: 74568941-74573626 and mmu_circ: chr8: 86399206-8639489 were validated among those upregulated in stroke compared to sham mice. Bioinformatics revealed a circRNA-miRNA-mRNA regulatory network comprising two circRNAs, ten miRNAs, and sixty-nine mRNAs. Pathway analysis showed that the differentially expressed circRNA (DECs) have multiple targets, such as inflammation and self-repair, remote non-ischemic injury or tissue remodeling after focal cerebral infarction and are involved in several signaling pathways, including PI3K-Akt, MAPK, Ras and Rap1, endocytosis, focal adhesion, and axon guidance [11,56] (Table 1).

In another study using the same model, it was shown that circular RNA derived from oxoglutarate dehydrogenase (circOGDH) physically interacts with miR-5112 in penumbra neurons of MCAO mice, enhancing the expression of COL4A4 to promote neuronal damage. On the other hand, knockdown of circOGDH significantly improved the viability of neuronal cells under ischemic conditions. Notably, circOGDH is more than 50 times upregulated in the plasma after acute IS (AIS), and its blood expression levels are positively correlated with the size of the ischemic penumbra in patients with AIS [12].

Growing evidence has demonstrated the regulatory role of circRNAs in neuroinflammation and neuronal apoptosis. Expression levels of circCDC14 in both plasma and the peri-infarct cortex of transient MCAO (tMCAO) mice are progressively increased at 12 h, and through to 24 h after reperfusion [13]. The downregulation of circCDC14A obtained though lentiviral vector reduced the activation of astrocytes in the peri-infarct cortex, mitigating brain damage during AIS. Furthermore, in HT22 cells treated with oxygen and oxygen deprivation/reoxygenation (OGD/R), the knockdowns of circCDC14A suppressed the reduction of cell viability by negatively regulating miR-23a-3p [14].

Another study also demonstrated the negative correlation between circ_0000831 and miR-16-5p in the brain of tMCAO mice. In particular, circ_0000831 could bind to miR-16-5p to activate the AdipoR2/PPARγ axis, diminishing neuroinflammation in cerebral ischemia [15].

Compared to the sham group, circCCDC9 levels were significantly reduced in MCAO mice at 24 h after ischemic injury and remained stable at 72 h. CircCCDC9 overexpression promoted BBB protection after ischemia/reperfusion (I/R) brain injury in mice by inhibiting apoptosis through post-transcriptional regulation of Notch1, NICD, and Hes1 expression [16]. Zhang et al. [17] found that the expression levels of circLIFR and kinase insert domain receptor (KDR) were downregulated in arterial wall tissues and ASMCs of IS patients. On the contrary, the forced expression of circLIFR was able to improve proliferation, migration, invasion, and apoptosis of human umbilical artery smooth muscle cells (HUASMC) through the miR1299/KDR axis.

The upregulation of circRNAs also plays a neuroprotective role against I/R damage by negatively regulating the expression of target genes [57]. MiRNA-143-3p overexpression contributed to autophagy and astrocyte apoptosis, as well as brain damage in MCAO rats. In astrocytes, circ_0025984 acts as a miR-143-3p sponge, which directly targets TET1 and diminishes its expression. TET1 overexpression significantly reduced astrocyte apoptosis and autophagy, as well as brain damage and neuron loss in mice with MCAO [18]. In another study, higher expression of ipsilateral miR-143 was observed in mice undergoing tMCAO than in sham-treated mice. Consistently, the expression levels of circDLGAP4 decreased in the ipsilateral hemispheres of tMCAO mice 24 h after reperfusion [19]. Bioinformatic analysis showed that circDLGAP4 contains a miR-143 target site and could act as a sponge to inhibit its activity [19].

Treatment with circUCK2 was associated with a substantial decrease in stroke volume in MCAO mice, along with an attenuation of BBB damage through the inhibition of endothelial to mesenchymal transition (EndoMT). CircUCK2 reduced cell apoptosis induced by oxygen glucose deprivation (OGD) by regulating transforming growth factor β (TGF-β)/mothers against signaling of decaptaplegic homolog 3 (Smad3). Additionally, circUCK2 acted as an endogenous miR-125b-5p sponge to inhibit miR-125b-5p activity, resulting in increased growth differentiation factor 11 (GDF11) expression and subsequent improvement in neuronal damage [20].

Yang et al. [21] identified a novel role of the circTTC3/miR-372-3p/TLR4 axis in cerebral ischemia-reperfusion injury (CIR) in a middle cerebral artery occlusion/repression (MCAO/R) mice model. CircTTC3 was significantly elevated in MCAO/R mice in vivo and in astrocytes treated with OGD. The increase in expression of circTTC3 was able to repress the proliferation and differentiation of neural stem cells (NSCs). In particular, circTTC3 was able to sponge miR-372-3p, resulting in an overexpression of TLR4 with consequences of a reversal of OGD lesion of astrocytes mediated by a depletion of circTTC3 and regulation of NSC.

Mehta et al. [22] identified the circRNA expression profile in the penumbral cortex of MCAO mice at 6 h, 12 h, and 24 h of reperfusion. They found that 238 circRNAs were modulated (change fold > 2) in at least one of the reperfusion time points, involved in metabolism, cell communication and binding with proteins, ions, and nucleic acids. Of those, 16 circRNAs contained binding sites for many miRNAs. It was observed that >80% of stroke-reactive circRNAs showed >60 miRNA binding sites each. The circ_016423 showed the maximum of 625 miRNA binding sites, with affinity for 521 different miRNAs.

Wang et al. [23] analyzed the post-stroke circRNAs expression profile by sequencing, identifying a total of 24,858 circRNAs, of which only 18.74% were known circRNAs. Compared to the sham group, there were 158 up- and 136 down-regulated circRNAs in the rat cortex after MCAO. Cluster, GO, and KEGG analyses suggested their involvement in numerous biological processes and pathways closely related to stroke. CircRNA-miRNA-mRNA interaction analysis revealed 577 sponge miRNA and 696 target mRNA [24]. Bioinformatics revealed that, among those, 15 circRNAs are involved in oxidative stress, apoptosis, inflammation, and nerve regeneration [23].

Chen et al. demonstrated the involvement of the circHIPK3/miR-148b-3p/CDK5R1/SIRT1 axis in the development of IS in a model of tMCAO. In particular, knockdown of circHIPK3 considerably reduced infarct volume, with improved cerebral microvascular endothelial cell (BMEC) apoptosis and mitochondrial dysfunction in tMCAO mice. Mechanistically, circHIPK3 acts as an endogenous sponge of miR-148b-3p to decrease its activity, resulting in upregulation of CDK5R1 and CDK5 expression and downregulation of SIRT1 expression, and subsequent BMEC apoptosis and mitochondrial dysfunction. From the obtained results, it is possible to propose circHIPK3 as a possible therapeutic approach to prevent apoptosis and mitochondrial dysfunction in IS patients [25].

In a previous study, approximately 395 circRNAs were identified by genome-wide RNA-Seq analysis of subcortical regions of the rat brain (tMCAO) [26]. It was found that about one-third of the DECs came from genes the mRNA levels of which also changed 24 h after tMCAO. DECs were involved in multiple processes, including the hippo signaling pathway, extracellular matrix-receptor interaction, and fatty acid metabolism. Most of the predicted circRNA-miRNA interactions exhibited functional roles in terms of regulating their target gene expression in the brain [27]. Kang et al. [28], using the GSE6088 database, found 467 increased and 490 decreased differential expressed genes (DEG) in atherosclerosis. Bioinformatics analysis demonstrated the involvement of DEGs with multiple cellular pathways, including glyoxylate, dicarboxylate, tyrosine, tryptophan, beta-alanine starch and sucrose metabolism, and fatty acid biosynthesis. An important regulatory role in atherosclerosis is played by circHIPK3, the knockdown of which can suppress both proliferation and apoptosis of VSMCs through the miR637/CDK6 axis [28].

Similar to other non-coding RNAs [58,59,60], circRNAs can be packed in exosomes, also involved in the pathogenesis of stroke. Wen et al. [29] evaluated the effect of circRNA in serum exosomes (serum-Exos) of patients with stable plaque atherosclerosis (SA) and unstable/vulnerable plaque atherosclerosis (UA). The results indicated that after treatment with serum-Exos UA, the expression of circRNA_0006896 in human umbilical vein endothelial cells (HUVECs) was upregulated, accompanied by a downregulated expression of miR1264 and an upregulated expression of DNA methyltransferase 1 (DNMT1) mRNA. Recently, DNMT1 has been identified as the direct target of miR1264, which can bind to the 3’UTR of the DNMT1 transcript to inhibit its expression [29]. The increased expression of circRNA_0006896 in serum-Exos from patients with UA promotes the proliferation and migration of HUVEC by negatively regulating the expression of miR-1264. In turn, this leads to an increase in DNMT1 expression and STAT3 phosphorylation, and a reduction in SOCS3 expression. The downregulation of SOCS3 results in the loss of its inhibitory effect on the JNK/STAT3 pathway, influencing the formation of vulnerable plaques and contributing to complex regulatory networks in patients with UA.

He et al. [30] investigated the interaction of fluoxetine-modulated circRNA and mRNA in MCAO rats. In total, 958 circRNA and 838 mRNA were differentially expressed. In particular, the expression of circMap2k1, which has a direct binding capacity to miR-135b-5p, was significantly increased in the MCAO group, but was suppressed after treatment with fluoxetine, resulting in a reduction of the infarcted area.

Chen et al. [31] identified the tanshinone IIA (TAN)-regulated network of ceRNA and elucidated the non-coding RNA (ncRNA) profile and signal pathways to attenuate atheroscleroris (AS). ApoE −/− mice were fed with a high-fat diet for 12 weeks to induce atherosclerosis, and half of the mice were treated with TAN. A total of 22 long non-coding RNAs (lncRNA), 74 microRNAs (miRNA), 13 circular RNAs (circRNA), and 1359 mRNAs in the AS plaque were identified from the sequencing analysis that was more significantly regulated by TAN mice. The pathways most significantly regulated by TAN were associated with inflammation and involved in signaling pathways of Ras, Rap1, MAPK, cAMP, and T cell receptor. Furthermore, the competitive network of competitive endogenous RNA (ceRNA) was built with main nodes, including circ-Tns3/let-7d-5p/Ctsl, circ-Wdr91/miR-378a-5p/Msr1, and circ-Cd84/miR- 30c/TLr2.

Wang et al. [32] investigated the role of circRNAs in neuronal stem cells differentiation; the treatment of IS remains unknown. NSCs were transducted with circHIPK2 siRNA (si-circHIPK2-NSCs) or vehicle control (si-circCon-NSCs) and microinjected into the lateral ventricle of the brain in 7-d post-tMCAO mice (Figure 2). In vitro, silencing of circHIPK2 facilitated NSCs directionally differentiated to neurons, but had no effect on the differentiation to astrocytes. In vivo, microinjected NSCs could migrate to the ischemic hemisphere after stroke induction. Si-circHIPK2-NSCs increased neuronal plasticity in the ischemic brain, conferred long-lasting neuroprotection, and significantly reduced functional deficits. Si-circHIPK2 regulates NSC differentiation, and microinjection of si-circHIPK2-NSCs exhibits a promising therapeutic strategy for neuroprotection and functional recovery after stroke. The experimental model adopted and main study results are depicted in Figure 2 as an example of the usefulness of the MCAO model.

## 4. OGD-Induced Neuron Injury

In vitro models are widely used to study and understand the molecular mechanisms involved in ischemic pathology. To improve the reliability and reproducibility of hypoxic/ischemic conditions in vitro, neuronal cell cultures are subjected to ischemic oxygen/glucose deprivation conditions (OGD/R). Ischemia-reperfusion (IR) injury significantly contributes to the morbidity and mortality associated with ischemic strokes. The IR lesion is secondary to the restoration of blood flow, after a prolonged period of ischemia, with a consequent supply of nutrients and oxygen to the tissue, which eventually lead to the activation of the inflammatory process and oxidative stress. Currently, several in vitro models are used to study and understand the molecular mechanisms involved in IR pathology. Neuronal cell cultures are subjected to ischemic oxygen/glucose deprivation (OGD/R) conditions to improve the reliability and reproducibility of hypoxic/ischemic conditions in vitro. The OGD model is obtained by replacing cell growth medium containing 4.5 g/L glucose under 95% sterile air/5% CO_2_ conditions with glucose-free Dulbecco’s Modified Eagle’s Medium (DMEM) for 6 h in a hypoxic incubator. After exposure to OGD, cells are oxygenated and maintained in regular medium for various periods of time. OGD/R resistance allows the study of cell proliferation, apoptosis, inflammation, and oxidative stress damage in neuronal cells. The experimental model of OGD, along with examples of its adoption in actual research studies, is depicted in Figure 3.

Using this model, Dai et al. [33] explored the role of several circ-RNAs (circ_0000647, CircUCK2, CircHECTD1, Circ_0072309) in SK-N-SH cells. The results showed that circUCK2 overexpression repressed infarct volumes, attenuated neuronal damage, and improved neurological deficits, and OGD-stimulated apoptosis contained in IS through regulation of miR-125b-5p/GDF11. In OGD- induced HT22 cells, HECTD1 silencing activated a neuroprotective effect by inhibiting neuronal cell apoptosis by responding to miR-133b/TRAF3 regulation [34]. Circ_0072309 increased OGD-induced cell survival and reduced apoptosis in IS by sponging miR-100 [35] (Table 1). Specifically, circ_0000647 was found to inhibit OGD/R-mediated SK-N-SH cell proliferation and promoted apoptosis, inflammation, and oxidative stress damage by reducing TRAF3 gene expression by targeting miR-126-5p, largely known for its involvement in multiple vascular diseases [61,62,63,64,65,66], thus promoting IS progression [33]. In the same cell model, Pei et al. [36] observed that the downregulation of circ_0101874 alleviated OGD-induced injury in SK-N-SH cells. In particular, circ_0101874 functioned as a sponge to inhibit miR-335-5p expression and improved PDE4D expression, attenuating neuronal apoptosis, inflammation, and oxidative stress in the development of ischemic stroke. These results suggest that the inhibition of circ_0101874 could be a promising target for IS treatment.

Angiogenesis plays an important role in vascular angiogenic remodeling and neurofunctional recovery after stroke. Growing evidence has shown that circRNAs are essential mediators of both vascular endothelial cell biology and angiogenesis. Therefore, the role of different circRNAs in cell damage induced by oxygen-glucose deprivation in HBMECs was investigated.

CircFUNDC1 expression was increased in the peripheral blood of patients with IS and human brain microvascular endothelial cells (HBMEC) treated with OGD. CircFUNDC1 knockdown relieved OGD-induced cell apoptosis and promoted OGD-blocked cell viability, HBMEC migration, and angiogenesis by inhibiting PTEN by miR-375 overexpression [37]. Following bioinformatics analyses predicted miR-222-3p as the target of circ_0006768 [38]. The expression of miR-222-3p is upregulated in the plasma of patients with IS compared to normal controls, and was negatively correlated with the expression of circ_0006768. Again, the angiogenesis capacity and cell viability improved following the inhibition of miR-222-3p in OGD/R-induced HBMECs transfected with anti-miR-222-3p [38]. The upregulation of circ_0006768 attenuates the lesions of the microvascular endothelial cells of the human brain induced by OGD/R through the upregulation of VEZF1 through the inhibition of miR-222-3p. The overexpression of circ0006768 in OGD-induced HBMECs promoted the recovery of OGD/R-inhibited angiogenesis and HBMEC migration capabilities, suggesting that circ0006768 could represent another possible therapeutic target in IS [38].

Using the same brain injury model of IS induced by oxygen-glucose deprivation/re-oxygenation (OGD/R), Liu et al. [39] evaluated the levels of circ0007865, miR-214-3p, and FK506-binding protein 5 (FKBP5) by quantitative real-time PCR in human brain microvascular endothelial cells (HBMECs). From the results, the expression of circ_0007865 and FKBP5 increased and miR-214-3p was reduced in OGD-treated HBMECs. Furthermore, silencing of circ_0007865 could promote cell proliferative angiogenesis and migration, and inhibit apoptosis in OGD-activated HBMECs in vitro. Mechanically, circ_0007865 could attenuate OGD-induced HBMEC damage by modulating the miR-214-3p/FKBP5 axis, suggesting a further promising therapeutic target [39].

The role of circ-camk4 in cell survival and apoptosis was investigated [40] using a model of Sprague Dawley (SD) rat middle cerebral artery occlusion (MCAO) following I/R brain damage. A significant increase in the expression of circ-camk4 was observed by approximately two times in the MCAO samples (*n* = 4) compared to that observed for the control group samples (*n* = 4). Subsequently, the primary culture of OGD/R-exposed neuronal cells was also used as an in vitro model to test whether circ-camk4 expression in neuronal cells responded to OGD for 3 or 6 h and to subsequent recovery. Circ-camk4 expression was upregulated at least two-fold in both OGD/R-3h and OGD/R-6h groups compared to the control group. Furthermore, a more pronounced amount of cell death was observed in SH-SY5Y cells exposed to OGD/R treatment. These results indicate that circ-camk4 expression is influenced by ischemia in neuronal cells, and plays a role in the pathogenesis of brain ischemic injury.

Further studies aimed to explore the functions of specific circRNAs (circDLGAP4, circHECTD1) and their interaction networks under the conditions of ischemic OGD [41,42]. Qui et al. [41] detected the expression levels of circDLGAP4, miR-503-3p, and neuronal growth regulator 1 (NEGR1) in HCN-2 cells treated with OGD. The level of CircDLGAP4 was reduced in HCN-2 cells after treatment with OGD. CircDLGAP4 overexpression promoted cell viability and suppressed cell death and inflammatory cytokine concentrations in OGD-treated HCN-2 cells. CircDLGAP4 acted as a sponge for miR-503-3p and the impacts of circDLGAP4 overexpression on cell viability, death, and inflammation in OGD-treated HCN-2 cells were reversed by miR-503-3p elevation. Furthermore, NEGR1 was the target gene of miR-503-3p. Inhibition of MiR-503-3p ameliorated OGD-induced HCN-2 cellular impairment, with NEGR1 knockdown abolishing this effect. The conclusions of this study established that CircDLGAP4 reduced OGD-induced HCN-2 cell damage by regulating the miR-503-3p/NEGR1 axis [41].

The stimulation of OGD-induced neuronal apoptosis promoted lactate dehydrogenase (LDH) release and increased inflammation in HCN-2 cells, which were all reversed by circ0007290 knockdown by adjusting the miR-496/PDCD4 axis [43]. The role and mechanism of circHECTD1 in OGD/R-induced cell damage have been investigated by detecting levels of CircHECTD1, miR-27a-3p, and Follistatin-like 1 (FSTL1) by quantitative real-time polymerase chain reaction (RT-qPCR) in a model of brain ischemia [42]. Circ_HECTD1 and FSTL1 were highly expressed and miR-27a-3p was decreased in OGD/R-treated HT22 cells. The knockdown of circHECTD1 could increase cell proliferative capacity and repress apoptosis in OGD/R-activated HT22 cells in vitro. The mechanical analysis found that circHECTD1 could mitigate OGD-induced cell damage by modulating the miR-27a-3p/FSTL1 axis [42]. Furthermore, in mouse stroke models with transient MCAO, the knockdown of circHECTD1 expression significantly reduced the infarct area, attenuated neuronal deficits with subsequent inhibition of astrocyte activation via macroautophagy/autophagy pathway [44]. The expression profile of a PHC3-derived circRNA, hsa_circ0001360 in HBMEC, was also investigated after exposure to OGD using RNA-seq technology. A total of 3978 circRNAs were detected among non-OGD and OGD-treated cells. In total, 74 circRNAs from OGD and non-OGD, including 14 up-regulated and 60 down-regulated circRNAs, were included; 14 circRNAs were chosen for further studies and compared to the non-OGD group, finding a significant increase in expression levels of this circRNA in HBMECs. Inhibition of circPHC3 significantly suppressed cell apoptosis and could bind to seven miRNAs. In particular, a relationship was identified between circPHC3 and miR-455-5p that suppresses the death of neuronal cells in IS and plays a protective role in stroke. Notably, circPHC3 promoted cell death and apoptosis through the sponging of miR455-5p to activate the expression of TRAF3 [45].

A total of 217 differentially expressed circRNAs were identified between ischemic brain tissues and normal controls. Among them, circGLIS3 has been shown as the common regulator of brain and retinal neurodegeneration. Silencing of circGLIS3 relieved ischemia-induced retinal neurodegeneration and MCAO-induced brain neurodegeneration in vivo. circGLIS3 silencing protected against OGD/R-induced retinal ganglion cells (RGC) damage in vitro. CircGLIS3 regulated neuronal cell injury by acting as a miR-203 sponge and its level was controlled by EIF4A3 [46].

Other studies evaluated the neuroprotective effects of two circRNAs: circSHOC2 and circ-Rps5. Chen et al. [47] used exosomes derived from ischemic-preconditioned astrocytes (IPAS-EXO) as models against IS starting from an ischemic model based on OGD in vitro and exosomes isolated from astrocytes. A significant increase in the expression of circSHOC2 was observed in the exosomes released by IPAS-CM, with similar results also observed in the mouse model of MCAO. The overexpression of circSHOC2 produced neuroprotective effects by reducing apoptosis and regulating autophagy. Furthermore, circSHOC2 regulates SIRT1 expression by sponging miR-7670-3p. Transfection with a siRNA-7670-3p and incubation with circSHOC2 extracellular vesicles attenuated ischemia-induced neuronal apoptosis in vivo and in vitro, while silencing SIRT1 reversed the protective effects of exosomal circSHOC2 on hypoxic brain neurons. Therefore, these results indicate that circSHOC2 improves neuronal damage by acting on the miR-7670-3p/SIRT1 axis, which could serve as a stroke therapeutic target.

Yang et al. [48] demonstrated that hypoxic pre-treated ADSC exosomes attenuated acute ischemic stroke-induced brain damage through circ-Rps5 administration by promoting M2 microglia/macrophage polarization. Bioinformatics analysis showed that circ-Rps5 interacts with miR-124-3p and SIRT7. Circ-Rps5 overexpression promotes SIRT7 expression by sponging miR-124-3p with a neuroprotective effect. SIRT7 is the oxidized-dependent nicotinamide adenine dinucleotide deacetylase, implicated in various procedures such as DNA damage repair, cellular signal transduction, and aging, and suppresses lipopolysaccharide (LPS)-induced inflammation and apoptosis via the NF-Κb pathway. Silencing of SIRT7 or overexpression of miR-124-3p reversed the protective effect of circ-Rps5 on LPS and MCAO-induced inflammatory cytokine expression by shifting microglia from M1 to M2 phenotype. The circRps5-modified ADSC exosome improved cognitive function by decreasing neuronal damage and shifting microglia phenotype to M2 in the hippocampus.

Yang et al. [49] explored the dysregulation of circulating circPHKA2 in patients with AIS, further confirming the regulatory mechanism of ceRNA of circPHKA2 in OGD-induced neurovascular damage in HBMEC. Lower circPHKA2 expression was observed in the blood of AIS patients and suppression of circPHKA2 re-expression in the OGD-induced brain IS cellular model in HBMEC. CircPHKA2 could sponge miR-574-5p to regulate SOD2 expression in HBMEC. Re-expression of circPHKA2 in OGD-induced HBMEC promoted the enhancement of cell proliferation and migration, as well as neovascularization and mitigated apoptosis, oxidative stress, and ER stress, indicating its neurovascular protection role in stroke.

Interestingly, the beneficial and protective effect was partially abrogated by inhibiting SOD2, or by storming miR-574-5p. The downregulation of SOD2 and the upregulation of miR-574-5p were paralleled by the downregulation of circPHKA2 in the blood of AIS patients; furthermore, a linear correlation was found between the expression of SOD2 and circPHKA2. Together with direct interactions between miR-574-5p and circPHKA2 or SOD2, these data revealed a circPHKA2-miR-574-5p-SOD2 ceRNA axis in stroke-induced neurovascular injury. Furthermore, circPHKA2 protected HBMEC from OGD-induced neurovascular injury by controlling SOD2 via miR-574-5p sponging. All these data reinforce the potential role of this axis as an antioxidant-based neurovascular protective strategy in IS [49].

Tang et al. [50] conducted in vitro and in vivo experiments to evaluate the role of Map2k6 in stroke injury using both transient cerebral artery occlusion in mice and OGD/R in HT22 cells to simulate I/R damage. Results showed a significant increase in circ_016719 and Map2k6 expression levels, with reduced miR-29c levels in both models. In HT22 cells, the knockdown of circ_016179 significantly increased miR-29c expression and cell proliferation, but reduced Map2k6 expression and cell apoptosis. Map2k6 has been identified as a direct target of miR-29c, which in turn could be deleted by circ_016179. This suggests that circ_016719 targets miR-29c, and regulates the expression and functions of Map2k6, which significantly contributes to the pro-apoptotic role of circ_016719.

In a further study, rabies virus extracellular glycoprotein-circSCMH1-extracellular vesicles were generated to selectively deliver circSCMH1 to the brains of animal models. CircSCMH1 levels were significantly reduced in the plasma of AIS patients, offering significant power in predicting stroke outcomes. Reduced levels of circSCMH1 were further confirmed in the plasma and peri-infarct cortex of photothrombotic stroke mice. Treatment with circSCMH1 improved functional recovery after stroke in animal models (mice and monkeys), and circSCMH1 improved neuronal plasticity and inhibited glial activation and infiltration of peripheral immune cells. CircSCMH1 binds to the transcription factor MeCP2 (methyl-CpG binding protein 2), promoting transcription and repression of the target gene MeCP2. MeCP2 is key in neuronal function, transcriptionally regulating the expression levels of genes with an established role in synaptic homeostasis and other functional brain processes, including brain-derived neurotrophic factor [51].

The role of circTLK1 in neurological damage, viability, and cell apoptosis was investigated by Wu et al. [52] in cellular models of OGD/R. Both circTLK1 and PTEN were highly expressed, while miR-26a-5p was under-expressed in this IS models. CircTLK1 knockdown reduced infarct volume and neurological damage in MCAO mouse models, and alleviated OGD/R-induced neuronal damage in vitro by adjusting the miR-26a-5p/PTEN/IGF-1 R/GLUT1 axis. Moreover, circ_TLK1 knockdown attenuated OGD/R-induced damage in rat adrenal pheochromocytoma cells (PC12) through the regulation of miR-136-5p. Circ_TLK1 downregulation was abolished by miR 335-3 sponging in cell damage induced by OGD/R [53].

The cell models used for the study of IS included HCMECs treated with oxygen and OGD/R. He et al. [54] found that silencing circHECTD1 significantly reversed OGD/R-induced promotion of HCMEC tube formation and migration, and significantly alleviated the EndoMT process in HCMECs through the mediation of the miR-335/NOTCH2 axis. Ren et al. [55] used the same OGD model in human brain microvascular endothelial cells (HBMVEC), finding Circ-Memo1 and SOS1 increased expressions and reduced miR-17-5p expression. Circ-Memo1 silencing promoted cell viability, inhibited ERK/NF-ĸB signaling pathway activation, reduced oxidative stress and inflammatory response, and inhibited cell apoptosis by regulating the miR-17 -5p/SOS1 axis.

## 5. Biological Effects of circRNAs in IS, Potential Therapeutic Targets, and Future Perspectives

Circular RNAs can exert their biological role in different ways, many of which are still under study. However, the specific molecules identified in this systematic review mostly act by interacting with specific molecules involved in the modulation of gene expression (e.g., microRNAs) or in specific biological pathways, as depicted in Figure 4. For example, the circUCK2 targets the miR-125-5b and the GDF11 to inhibit neuronal damage. Hence, an augmentation of circUCK2 levels could eventually represent a potential therapeutic strategy to limit the extension of the cerebral infarcted area [20] (Table 2). On the contrary, silencing of the circHIPK2 in neuronal stem cells is potentially able to increase neuronal plasticity and to foster neuroprotection [32] (Figure 2). The most relevant potential therapeutic impact of the circRNAs described in this systematic review are listed in Table 2.

CircRNAs and other non-coding RNAs have the potential to interact with biological pathways in multiple ways. Current available evidence is mostly focused on relatively simple biological interactions, suggesting that we have just scratched the surface of a much larger body of evidence. An interesting class of non-coding RNAs, whose association with neurodevelopment and neurodegenerative disorders has been recently described, are PiWi-interacting RNA (piRNA), small non-coding RNAs that interact with PiWi-subtypes of the Argonaute protein complex [67]. The new scientific evidence linking multiple classes of non-coding RNAs to IS pathophysiology, along with the continuous advancements in therapeutic strategies aiming at interfering with their expression levels and/or function, open new avenues to the development of further and more specific therapeutic targets [68]. 

## 6. Conclusions

IS is one of the leading causes of death and disability worldwide. Studying the functions and mechanisms of circRNAs in biological systems under normal and pathological conditions can lead to potential opportunities to identify biomarkers and novel therapeutic targets for IS. Taken together, circRNAs play an important role in cerebral ischemia by modulating cell survival, the inflammatory process, angiogenesis, oxidative stress, polarization of microglia, neuronal apoptosis, and BBB permeability. The most common experimental models for stroke research include MCAO mice and OGD/R in vitro models. These experimental models will continue to be useful in defining the role of circRNA in AIS, helping to better understand the mechanisms underlying ischemic brain damage. Many circRNAs are associated with various pathophysiological parameters in AIS, such as infarct volume, brain edema, stroke severity, and clinical outcome. Therefore, in the future, circRNAs might be crucial to better understand the biological mechanisms underlying the development of ischemic brain injury and its complex clinical pathophysiology. Furthermore, circRNA expression levels can be detected in the bloodstream [6], and could therefore act as non-invasive biomarkers of diagnosis and prognosis for patients with AIS.

## Figures and Tables

**Figure 1 biomolecules-13-00214-f001:**
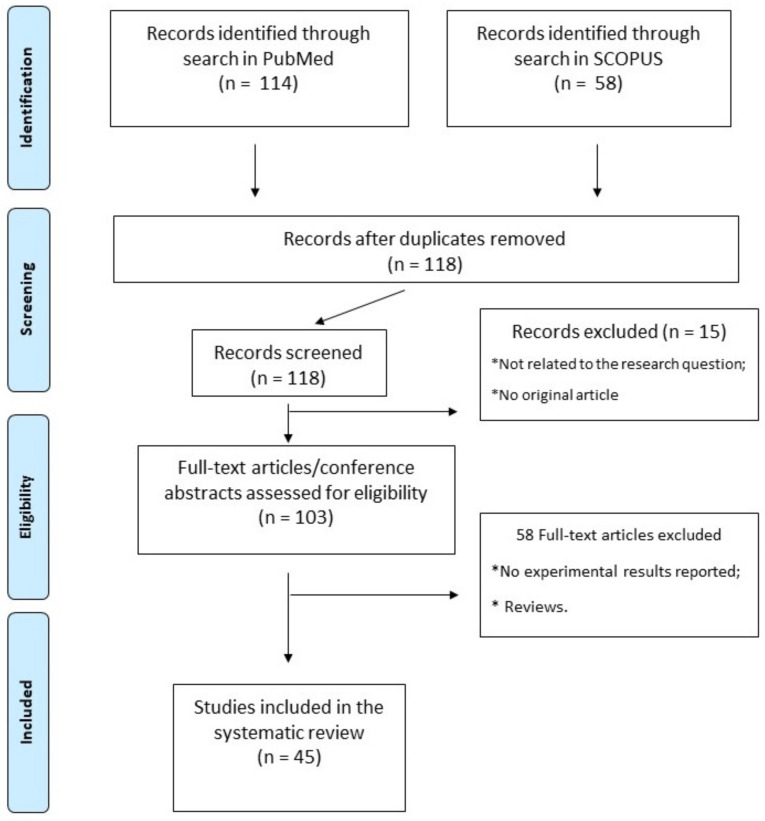
PRISMA flowchart. Article selection flowchart, according to PRISMA guidelines.

**Figure 2 biomolecules-13-00214-f002:**
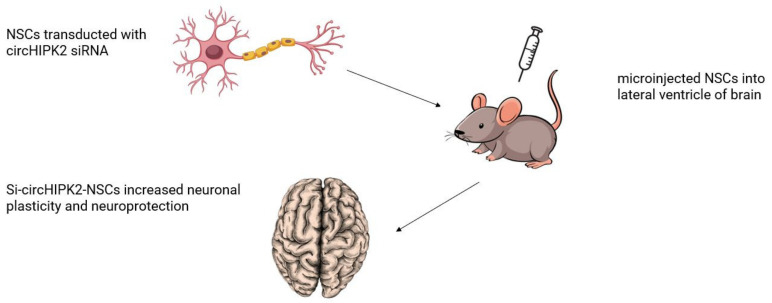
**Role of circRNAs in neural stem cells differentiation and in the treatment of IS.** NSCs were transducted with circHIPK2 siRNA and microinjected into lateral ventricle of brain at 7 days post tMCAO. Si-circHIPK2-NSCs regulates NSC differentiation and increased neuronal plasticity in the ischemic brain. NCS: neural stem cells; IS: ischemic stroke; circRNA: circular RNA; tMCAO: transient middle cerebral arterial occlusion; siRNA: small interfering RNA.

**Figure 3 biomolecules-13-00214-f003:**
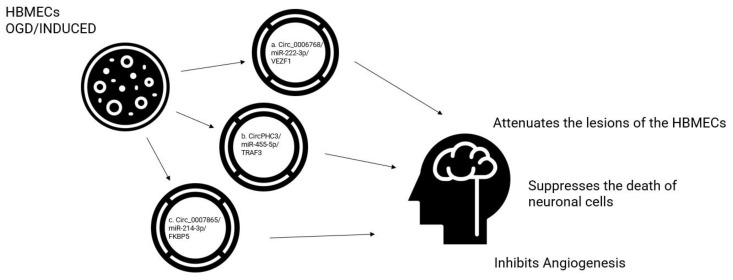
**The effect of circRNAs in OGD-treated HMBECs.** (a) Upregulated circ_0006768 attenuates the lesion of HBMEC OGD induced through the upregulation of VEZF1 and the inhibition of miR-222-3p. (b) Inhibition of circPHC3 suppresses the death of neuronal cells by sponging of miR-455-5p to activate the expression of TRAF3. (c) Overexpression of circ0007865 promotes the inhibition of angiogenesis by modulating the miR-214-2p/FKBP5 axis. circRNA: circular RNA; HMBEC: human brain microvascular endothelial cells; miR: microRNA; VEZF1: vascular endothelial zinc finger 1; FKBP5: FK506-binding protein 5; TRAF3: TNF receptor associated factor 3.

**Figure 4 biomolecules-13-00214-f004:**
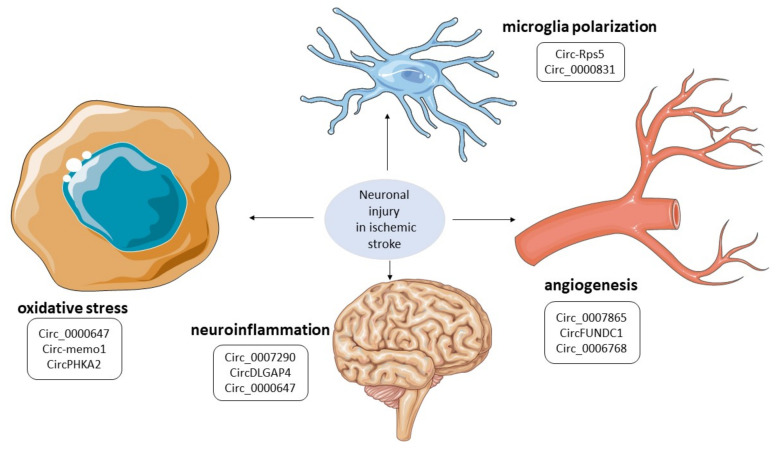
**CircRNAs involved in key biological pathways of ischemic brain injury.** CircRNAs are involved in several biological pathways that contribute to the pathophysiology of ischemic stroke by mediating pathophysiological mechanisms, including angiogenesis, neuroinflammation, microglial activation, and oxidative stress.

**Table 1 biomolecules-13-00214-t001:** Lists of articles.

Authors	Topics	Experiment Models	Type of Study	Aim	Pathways
Li et al. [11]	Focal cortical infarction	Adult male C57BL/6J mice subjected to permanent distal middle cerebral artery occlusion (MCAO)	In vivo	Expression and roles of circRNAs in non-ischemic remote regions after ischemic stroke	Profile the circRNA expression in the mouse ipsilateral thalamus at 7 and 14 d after MCAO
Liu et al. [12]	Acute ischemic stroke (AIS)	Middle cerebral artery occlusion mice; the plasma of 45 patients with AIS	In vivo	Investigate whether the circOGDH is a potential biomarker for penumbra in patients with AIS and its role in ischemic neuronal damage	Sequestering of microRNA-5112 by circOGDH enhanced COL4A4 expression to elevate neuron damage
Zuo et al. [13]	Ischemic stroke (IS)	tMCAO mice	In vivo	Seek out the regulatory mechanism of circCDC14A in neuroinflammatory injury in tMCAO mice	The expression level of circCDC14A in the peri-infarct cortex and plasma of mice
Huo et al. [14]	Ischemic stroke (IS)	Middle cerebral artery occlusion (MCAO) model and oxygen and glucose deprivation/reoxygenation (OGD/R)-treated HT22 cells	In vivo/in vitro	Role of circCDC14A in cerebral ischemia-reperfusion (CI/R) injury in vivo e in vitro	CircCDC14A acted as a sponge for miR-23a-3p and promoted the expression of chemokine stromal-derived factor-1 (CXCL12) by negatively regulating miR23a-3p
Huang et al. [15]	Ischemic stroke (IS)	Middle cerebral artery occlusion (MCAO) model	In vivo	Role of circ_0000831 in the pathogenesis of stroke	circ_0000831 bound to miR-16-5p and downregulated miR-16-5p, and AdipoR2 was targeted by miR-16-5p and increased PPARγ expression in microglia.
Wu et al. [16]	Ischemic stroke (IS)	tMCAO mice model	In vivo	Role of circCCDC9 in the pathogenesis of stroke	Overexpression of circCCDC9 inhibited the expression of Caspase-3, Bax/Bcl-2 ratio, and the expression of Notch1, NICD, and Hes1 in tMCAO mice.
Zhang et al. [17]	Intracranial aneurysm (IA)	Human umbilical artery smooth muscle cells (HUASMCs)	In vitro	Role and mechanism of circRNA LIF receptor subunit alpha (circLIFR, circ_0072309)	CircLIFR directly targeted miR-1299, and miR-1299 as a downstream mediator of circLIFR in regulating the proliferation, migration, invasion, and apoptosis of HUASMCs
Zhou et al. [18]	Ischemic stroke (IS)	MCAO mice model	In vivo	Treatment with miR-143-3p inhibitor or circ_0025984 significantly decreased astrocyte apoptosis and autophagy, as well as cerebral injury and neuron loss	Circ_0025984 and TET1 as a sponge and target of miR-143-3p
Bai et al. [19]	Ischemic stroke (IS)	The plasma of acute ischemic stroke patients (13 females and 13 males); mouse stroke model	In vivo	Role of circRNAs in stroke	CircRNA DLGAP4 functions as an endogenous microRNA-143 sponge to inhibit miR-143 activity, resulting in the inhibition of homologous to the E6-AP C-terminal domain E3 ubiquitin-protein ligase 1 expression
Chen et al. [20]	Ischemic stroke (IS)	HT22 cells; MCAO model mice	In vivo/in vitro	Role of circRNA UCK2 in ischemic stroke-associated neuronal injury	CircUCK2/miR-125b-5p/GDF11 axis is an essential signaling pathway during ischemia stroke
Yang et al. [21]	Cerebral infarction	Middle cerebral artery occlusion/repression (MCAO/R) model in C57BL/6J mice; neural stem cell (NSCs)	In vivo/in vitro	Explore the impact of circTTC3 on CIR injury and NSCs	CircTTC3 regulates CIR injury and NSCs by the miR-372-3p/TLR4 axis in cerebral infarction
Mehta et al. [22]	Ischemic stroke (IS)	Male C57BL/6J mice subjected to transient middle cerebral artery occlusion	In vivo	Stroke changes the circRNAs expression profile in the mouse brain	Levels of 14 236 circRNAs in the cerebral cortex of adult mice as a function of reperfusion time after transient focal ischemia
Wang et al. [23]	Ischemic stroke (IS)	Middle cerebral artery occlusion (MCAO) in rats	In vivo	circRNAs participate in the complex regulatory networks involved in stroke pathogenesis	15 key potential circRNAs were predicted to be involved in the post-transcriptional regulation of a series of downstream target genes, which are widely implicated in post-stroke processes, such as oxidative stress, apoptosis, inflammatory response, and nerve regeneration, through the competing endogenous RNA mechanism
Duan et al. [24]	Ischemic stroke (IS)	MCAO in rats	In vivo	Explore the relationship between circRNAs and ischemic stroke induced by middle cerebral artery occlusion (MCAO) in rats	Expression profile of circRNA in brain tissues
Chen et al. [25]	Ischemic stroke (IS)	tMCAO	In vivo	the involvement of the circHIPK3/miR-148b-3p/CDK5R1/SIRT1 axis in the development of IS in a model of tMCAO	circHIPK3 functions as an endogenous sponge of miR-148b-3p to decrease its activity and subsequent apoptosis and mitochondrial dysfunction
Filippenkov et al. [26]	Ischemic stroke (IS)	tMCAO	In vivo	Regulation of neurotransmission in the rat brain after ischemia by the action of circRNAs	CircRNAs can persist as potential miRNA sponges for the protection of mRNAs of neurotransmitter gene
Dan Lu et al. [27]	Acute Ischemic stroke (AIS)	Blood samples from mice and patients; mice’s tissue	In vivo	Examine whether the blood-borne circRNA could be promising candidates as adjunctive diagnostic biomarkers and their pathophysiological roles after stroke	An increasing number of circRNA were significantly altered after different time points after IS. The circRNA-targeted gene was associated with the Hippo signaling pathway, extracellular matrix receptor interaction, and fatty acid metabolism. CircBBS2 and circPHKA2 were differentially expressed in the blood of AIS patients
Kang et al. [28]	Atherosclerosis	HUASMC cells	In vitro	Identify differently expressed mRNAs in atherosclerosis by analyzing the GSE6088 database	CircHIPK3 regulates the proliferation and apoptosis of VSMCs by influencing the miR-637/CDK6 axis
Wen et al. [29]	Atherosclerosis	Human umbilical vein endothelial cell (HUVEC); patients with SA or UA	In vitro	Evaluate the effect of circular RNA molecules in serum exosomes from patients with stable plaque atherosclerosis (SA) and unstable plaque atherosclerosis (UA)	circRNA_0006896, miR-1264-DNMT1 axis
He et al. [30]	Cerebral ischemic stroke	Middle cerebral artery occlusion (MCAO) rat models	In vivo	Investigate the functions of fluoxetine and identification of fluoxetine-mediated circRNAs and mRNAs in cerebral ischemic stroke	CircMap2k1/miR-135b-5p/Pidd1 axis involved in cerebral ischemic stroke
Chen et al. [31]	Atherosclerosis (AS)	ApoE-/-mice; RAW264.7 cells	In vivo/in vitro	Characterize ncRNA profile and signal pathways to attenuate AS	22 long non-coding RNAs, 74 microRNAs, 13 circular RNAs, and 1359 mRNA in AS plaque were more significantly regulated from TAN mice
Wang et al. [32]	Ischemic stroke (IS)	Middle cerebral artery occlusion (tMCAO) mice; NCSs were transducted with circHIPK2 siRNA	In vivo/in vitro	Role of circHIPK2 in neural stem cell (NSC) differentiation and the treatment of IS	Si-circHIPK2 regulates NSC differentiation, and microinjection of si-circHIPK2-NSCs exhibits a promising therapeutic strategy for neuroprotection and functional recovery after stroke
Dai et al. [33]	Ischemic stroke (IS)	Human neuroblastoma cell line (SK-N-SH)	In vitro	The function of circ_0000647 in the pathogenesis of IS	Level of circ0000647, microRNA-126-5p, and TNF receptor-associated factor 3 (TRAF3)
Dai et al. [34]	Ischemic stroke (IS)	Mouse middle cerebral artery occlusion (MCAO) model and oxygen-glucose deprivation (OGD) model in HT22 cells	In vivo/in vitro	Explored the functional role of circRNA-HECTD1 and its underlying mechanism in cerebral ischemia/reperfusion injury	Circ-HECTD1 knockdown inhibited the expression of TRAF3 by targeting miR-133b, thereby attenuating neuronal injury caused by cerebral ischemia
Zhao et al. [35]	Ischemic stroke (IS)	Serum of patients with IS; LIFR humanized mice with middle cerebral artery occlusion (MCAO)	In vivo	Investigate the expression of circ_0072309 in patients with ischemic stroke and LIFR humanized mice with MCAO	Circ_0072309- miR-100- mTOR regulatory axis could alleviate IS
Pei et al. [36]	Ischemic stroke (IS)	SK-N-SH cells with oxygen-glucose deprivation (OGD) treatment	In vitro	Role of circ_0101874 in the pathogenesis of IS	Circ_0101874 knockdown alleviated OGD-induced neuronal cell injury by suppressing PDE4D via regulating miR-335-5p
Bai et al. [37]	Ischemic stroke (IS)	Human brain microvascular endothelial cells (HBMECs)	In vitro	Investigate the role of circRNA FUN14 domain containing 1 (circFUNDC1) in oxygen-glucose deprivation (OGD)-treated HBMECs	Expression of circFUNDC1, microRNA-375, and phosphatase and tensin homolog (PTEN)
Li et al. [38]	Ischemic stroke (IS)	Human brain microvascular endothelial cells (HBMECs)	In vitro	Explore the function and functional mechanism of circ_0006768 in oxygen-glucose deprivation/reoxygenation (OGD/R)-induced brain injury models of ischemic stroke	Expression of circ0006768, microRNA-222-3p, and vascular endothelial zinc finger 1 (VEZF1)
Liu et al. [39]	Acute ischemic stroke (AIS)	Human brain microvascular endothelial cells (HBMECs)	In vitro	Explore the role and mechanism of circ_0007865 in the oxygen-glucose deprivation (OGD)-induced cell damage in AIS	Circ_0007865 acted as a sponge of miR-214-3p to regulate FKBP5
Zhang et al. [40]	Ischemic stroke (IS)	MCAO rats; neuron cells exposed to oxygen-glucose deprivation/reperfusion (OGD/R)	In vivo/in vitro	Examine the role of circRNAs in cerebral I/R injury	Pathways that involve circcamk4 included the glutamatergic synapse pathway, MAPK signaling pathway, and apoptosis signaling pathways, all of which are known to be involved in brain injury after I/R
Qui et al. [41]	Ischemic stroke (IS)	Human cortical neuronal cells-2 (HCN-2)	In vitro	Explore the functions and mechanisms of circRNA DLG-associated protein 4 (circDLGAP4) in IS development	Expression of circDLGAP4, microRNA-503-3p, and NEGR1 in OGD-induced IS cell model
Zhang et al. [42]	Cerebral infarction	Mouse hippocampal cells (HT22)	In vitro	Explore the role and mechanism of circ_HECTD1 in OGD/R-induced cell injury in cerebral ischemia	Circ_HECTD1/miR-27a-3p/FSTL1 axis
Wang et al. [43]	Ischemic stroke (IS)	HCN-2 cells	In vitro	Investigate the role and mechanism of circ_0007290 in ischemic stroke	Knockdown of circ_0007290 alleviated OGD-induced neuronal injury by regulating the miR-496/PDCD4 axis, providing a novel insight into the pathology of IS
Han et al. [44]	Ischemic stroke (IS)	tMCAO mice; plasma samples from AIS patients	In vivo	Role of circHECTD1 in stroke	CircHECTD1 functions as an endogenous miR-142 sponge to inhibit miR-142 activity, resulting in the inhibition of TIPARP
Xu et al. [45]	Ischemic stroke (IS)	Brain microvascular endothelial cells (BMECs)	In vitro	Investigate profile circRNAs in human BMECs after oxygen-glucose deprivation (OGD), and find promising biomarkers in ischemic stroke	CircPHC3 acted as a miR-455-5p sponge to activate TRAF3 to promote cell death and apoptosis in human BMECs after OGD
Jiang et al. [46]	Ischemic stroke (IS)	C57BL/6J mice were subjected to transient middle cerebral artery occlusion	In vivo	circRNAs as regulators and diagnostic markers for cerebral neurodegeneration and retinal neurodegeneration	cGLIS3 regulated neuronal cell injury by acting as a miR-203 sponge and its level was controlled by EIF4A3
Chen et al. [47]	Ischemic stroke (IS)	Middle cerebral artery occlusion (MCAO) mouse model; model based on oxygen-glucose deprivation (OGD) and isolated resultant exosomes from astrocytes	In vivo/in vitro	Investigate the neuroprotective roles and mechanisms of circSHOC2 in ischemic-preconditioned astrocyte-derived exosomes (IPAS-EXOs) against ischemic stroke	CircSHOC2 in IPAS-EXOs suppressed neuronal apoptosis and ameliorated neuronal damage by regulating autophagy and acting on the miR-7670-3p/SIRT1 axis
Yang et al. [48]	Ischemic stroke (IS)	Adipose-derived stem cells (ADSCs)	In vitro	CircRNA expression between exosomes and hypoxic pre-treated ADSC exosomes	Exosomes from hypoxic pre-treated ADSCs attenuated acute ischemic stroke-induced brain injury via delivery of circ-Rps5 and promoted M2 microglia/macrophage polarization
Yang et al. [49]	Acute ischemic stroke (AIS)	Human brain microvascular endothelial cells (HBMEC)	In vitro	Investigate the role and mechanism of circPHKA2 in oxygen-glucose-deprivation (OGD)-induced stoke model in human brain microvascular endothelial cells (HBMEC)	CircPHKA2 could protect HBMEC against OGD-induced cerebral stroke model via the miR-574-5p/SOD2 axis
Tang et al. [50]	Ischemic stroke (IS)	I/R injury models; HT22 cells	In vitro	The role played by Map2k6 in stroke injury and the mechanism of action	Expression of circ016719, microRNA-29c, and Map2k6, and roles in cell proliferation and apoptosis
Yang et al. [51]	Ischemic stroke (IS)	The plasma of patients with acute ischemic stroke; rodent and nonhuman primate IS model	In vivo	Role of circRNA in ischemic brain injury	CircSCMH1 mechanistically binds to the transcription factor MeCP2 (methyl-CpG binding protein 2), thereby releasing repression of MeCP2 target gene transcription
Wu et al. [52]	Ischemic stroke (IS)	Middle cerebral artery occlusion (MCAO) mouse models in vivo and oxygen-glucose deprivation and reoxygenation (OGD/R) cell models in vitro	In vivo/in vitro	Explore the mechanism of circTLK1 in IS	CircTLK1 knockdown relieved IS via the miR-26a-5p/PTEN/IGF-1 R/GLUT1 axis
Zhang et al. [53]	Ischemic stroke (IS)	Rat adrenal pheochromocytoma cell line (PC-12)	In vitro	Role of Circ_TLK1 in the pathogenesis of IS	Circ_TLK1 acts as miR-136-5p sponge promoting upregulation of FSTL resulting in activation of apoptosis in PC12 cells
He et al. [54]	Ischemic stroke (IS)	Human cerebral microvascular endothelial cells (HCMEC); MCAO mice	In vivo/in vitro	Role of circHECTD1 in IS	CircHECTD1 knockdown significantly alleviated the EndoMT process in HCMECs via the mediation of the miR-335/Notch2 axis
Ren et al. [55]	Ischemic stroke (IS)	Human brain microvascular endothelial cells (HBMVECs)	In vitro	Role of circ-Memo1 in cerebral hypoxia/reoxygenation	Relationships between circ-Memo1, miR-17-5p, and SOS1

**Table 2 biomolecules-13-00214-t002:** Therapeutic effects of circRNA in ischemic stroke.

CircRNA	Target	Impact of circRNA	Therapeutic Effects
**circUCK2**	miR-125-5b/GDF11	Inhibit neuronal damage	Reduce infarct volumes
**circHECTD1**	miR-133b/TRAF332	Inhibit apoptosis	Reduce infarct volumes and improve neurological deficits
**circFUNDC1**	miR-375/PTEN	Reduce neurological deficits	Promote the ability of migration and recovery of angiogenesis
**circDLGAP4**	miR-503-3p/NEGR1	Inhibit apoptosis and neuroinflammation	Promote cell viability
**circHECTD1**	miR-27a-3p/FSTL1	Inhibit oxidative stress	Mitigate neuronal damage by regulating autophagy
**circSHOC2**	miR-7670-3p/SIRT1	Inhibit apoptosis	Promote neuroprotective effects by regulating autophagy
**circGLIS3**	miR-203/EIF4A3	Inhibit neuroinflammation	Alleviate retinal neurodegeneration
**circPHKA2**	miR-574-5p/SOD2	Inhibit apoptosis and oxidative stress	Improve cell proliferation and neovascularization
**circ-Memo1**	miR-17-5p/SOS1	Inhibit oxidative stress	Reduce the inflammatory response
**circTLK1**	miR-26a-5p/PTEN	Inhibit apoptosis	Reduce infarction volume and neurological damage

## Data Availability

No new data were created or analyzed in this study. Data sharing is not applicable to this article.

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
