# Peer review of "Circular RNAs in Ischemic Stroke: Biological Role and Experimental Models"

_biomolecules, 2023, doi:10.3390/biom13020214_

Round 1
Reviewer 1 Report
The present review paper has comprehensively discussed about the role of circular RNAs in the pathophysiology of ischemic stroke. The paper is well written and covers most of the literature regarding this important topic in the field. The authors also represented an informative table in the manuscript. Overall, I think the paper has a good quality in the current format to be published by the journal. I have few suggestions that can help the paper more informative.
1. A representative figure/diagram would be helpful categorizing most important circulating RNAs that show which in which biological/posttranslational pathways they are involved in ischemic stroke.
2. Although authors on/off have mentioned about therapeutic effects of targeting circulating RNAs would be helpful in stroke, an additional section specifically discuss about this issue would be more informative and organized. Perhaps an extra table can help in this regard as well.
Author Response
The present review paper has comprehensively discussed about the role of circular RNAs in the pathophysiology of ischemic stroke. The paper is well written and covers most of the literature regarding this important topic in the field. The authors also represented an informative table in the manuscript. Overall, I think the paper has a good quality in the current format to be published by the journal. I have few suggestions that can help the paper more informative.
Authors’ answer: we thank the Reviewer for taking the time to revise our manuscript and for the positive feedback. We have revised the manuscript according to all comments/suggestions.
- A representative figure/diagram would be helpful categorizing most important circulating RNAs that show which in which biological/posttranslational pathways they are involved in ischemic stroke.
Authors’ answer: we added a further figure, showing the most important pathways involved:
(see attached file to view the figure).
- Although authors on/off have mentioned about therapeutic effects of targeting circulating RNAs would be helpful in stroke, an additional section specifically discuss about this issue would be more informative and organized. Perhaps an extra table can help in this regard as well.
Authors’ answer: we added a new paragraph to better discuss the potential role of the molecules discussed in the article as potential therapeutic targets and an additional table 2, highlighting the targets and potential therapeutic effects for each molecule:
“5. Biological effects of circRNAs in IS, potential therapeutic targets and future perspectives
Circular RNAs can exert their biological role in different ways, many of which are still object of study to better understand their fine mechanisms. However, the specific molecules identified in this systematic review mostly act by interacting with specific molecules involved in the modulation of gene expression (e.g. microRNAs) or in specific biological pathways, as depicted in Figure 4. For example, the circUCK2 targets the miR-125-5b and the GDF11 to inhibit neuronal damage. Hence, an augmentation of circUCK2 levels could eventually represent a potential therapeutic strategy to limit the extension of the cerebral infarction area 22 (table 2). On the contrary, silencing of the circHIPK2 in neuronal stem cells is potentially able to increase neuronal plasticity and to foster neuroprotection 37 (Figure 2). The most relevant potential therapeutic impact of the circRNAs described in this systematic review are listed in Table 2.
CircRNAs and other noncoding RNAs have the potential to interact with biological pathways in multiple ways. Currently available evidence is mostly focused on relatively simple biological interactions, suggesting that we have just been scratching the surface of a much larger body of evidence. An interesting class of noncoding RNAs whose associa-tion with neurodevelopment and neurodegenerative disorders has been recently described are PiWi-interacting RNA (piRNA), small noncoding RNAs that interact with PiWi-subtypes of the Argonaute protein complex 67 . Along with new scientific evidence linking multiple class of noncoding RNAs to the pathophysiology of IS, the continuous advancements in the development of therapeutic strategies to interfere with their expression levels and/or function opens new avenues to the development of novel and more specific therapeutic strategies 68 .
Table 2. Therapeutic effects of circRNA in ischemic stroke
|
CircRNA |
Target |
Function of circRNA |
Therapeutic Effects |
|
circUCK2 |
miR-125-5b/GDF11 |
Inhibit neuronal damage |
Reduce infarct volumes |
|
circHECTD1 |
miR-133b/TRAF332 |
Inhibit apoptosis |
Reduce infarct volumes and improve neurological deficits |
|
circFUNDC1 |
miR-375/PTEN |
Reduce neurological deficits |
Promote the ability of migration and recovery of angiogenesis |
|
circDLGAP4 |
miR-503-3p/NEGR1 |
Inhibit apoptosis and neuroinflammation |
Promote cell viability |
|
circHECTD1 |
miR-27a-3p/FSTL1 |
Inhibit oxidative stress |
Mitigate neuronal damage by regulating autophagy |
|
circSHOC2 |
miR-7670-3p/SIRT1 |
Inhibit apoptosis |
Promote neuroprotective effects by regulating autophagy |
|
circGLIS3 |
miR-203/EIF4A3 |
Inhibit neuroinflammation |
Alleviate retinal neurodegeneration |
|
circPHKA2 |
miR-574-5p/SOD2 |
Inhibit apoptosis and oxidative stress |
Improve cell proliferation and neovascularization |
|
circ-Memo1 |
miR-17-5p/SOS1 |
Inhibit oxidative stress |
Reduce the inflammatory response |
|
circTLK1 |
miR-26a-5p/PTEN |
Inhibit apoptosis |
Reduce infarction volume and neurological damage |
Reviewer 2 Report
Dear Authors,
The present manuscript is nicely written on this particular topic. This is one of the major issues of therapeutics with stroke and requires needful attention. In recent years, A lot of review and research has been already published on this topic and I would like you to please update your literature.
I would like to suggest you add a few more pieces of information from the below article to make it more meaningful and scientifically sound.
Few examples:
PiWi RNA in Neurodevelopment and Neurodegenerative Disorders
Coding and non-coding nucleotides': The future of stroke gene therapeutics
Circulatory MicroRNAs as Potential Biomarkers for Stroke Risk
Impact of CircRNAs on Ischemic Stroke
My other suggestion is please include more research data in a table which are used for different stroke models in different conditions as well as please add an In-Vitro Study data table that will be more useful for this manuscript.
Please revise your "circRNA expression in OGD-induced neuron injury" it's very vague and confuses the direction of this article. Please write clear and scientific terms.
Please revise your title as it should convey the heart of your manuscript story.
Please revise your abstract as per your manuscript as it seems it does not match.
Author Response
The present manuscript is nicely written on this particular topic. This is one of the major issues of therapeutics with stroke and requires needful attention. In recent years, A lot of review and research has been already published on this topic and I would like you to please update your literature.
Authors’ answer: we thank the Reviewer for taking the time to revise our manuscript and for the positive feedback. We have updated the manuscript with more recent scientific evidence.
I would like to suggest you add a few more pieces of information from the below article to make it more meaningful and scientifically sound.
Few examples:
PiWi RNA in Neurodevelopment and Neurodegenerative Disorders
Coding and non-coding nucleotides': The future of stroke gene therapeutics
Circulatory MicroRNAs as Potential Biomarkers for Stroke Risk
Impact of CircRNAs on Ischemic Stroke
Authors’ answer: we thank the Reviewer for this suggestion. We have updated the manuscript with the suggested topics, and the following additional references have been added to support the discussion:
- Chavda V, Madhwani K, Chaurasia B. PiWi RNA in Neurodevelopment and Neurodegenerative Disorders. Curr Mol Pharmacol. 2022;15(3):517-531. doi: 10.2174/1874467214666210629164535
- Chavda V, Madhwani K. Coding and non-coding nucleotides': The future of stroke gene therapeutics. Genomics. 2021 May;113(3):1291-1307. doi: 10.1016/j.ygeno.2021.03.003.
- Mens MMJ, Heshmatollah A, Fani L, Ikram MA, Ikram MK, Ghanbari M. Circulatory MicroRNAs as Potential Biomarkers for Stroke Risk: The Rotterdam Study. Stroke. 2021 Mar;52(3):945-953. doi: 10.1161/STROKEAHA.120.031543
- Liu M, Liu X, Zhou M, Guo S, Sun K. Impact of CircRNAs on Ischemic Stroke. Aging Dis. 2022 Apr 1;13(2):329-339. doi: 10.14336/AD.2021.1113.
My other suggestion is please include more research data in a table which are used for different stroke models in different conditions as well as please add an In-Vitro Study data table that will be more useful for this manuscript.
Authors’ answer: we thank the Reviewer for this suggestion. Key research results were already presented in table 1. We have now updated the table in the manuscript, also adding a column indicating the type of study (in vitro or in vivo).
Please revise your "circRNA expression in OGD-induced neuron injury" it's very vague and confuses the direction of this article. Please write clear and scientific terms.
Authors’ answer: we have revised the title of paragraph 4 into “OGD-induced neuron injury” to avoid confusion and to conform it with the title used for paragraph 3.
Please revise your title as it should convey the heart of your manuscript story.
Authors’ answer: we have revised the title as suggested, into “Circular RNAs in ischemic stroke: biological role and experimental models”.
Please revise your abstract as per your manuscript as it seems it does not match.
Authors’ answer: we have revised the abstract as suggested:
“Ischemic stroke is among the leading causes for morbidity, disability, and mortality world-wide. Despite the recent progress in the interventional treatment of acute ischemic stroke, timely intervention is still a challenge. Hence, strategies to counteract ischemic brain injury during and around the acute event are still lagging, also because the knowledge of the underlying mechanisms is still limited. In fact, despite the increasing understanding of the complex pathophysiology underlying ischemic brain injury, we are still lacking some relevant pieces of information, particularly regarding the fine modulation of biological processes. In this context, emerging evidence is showing that the modulation of circular RNAs, a class of highly conserved non-coding RNA with a closed-loop structure, are involved in pathophysiological processes be-hind ischemic stroke, unveiling a number of potential therapeutic targets and possible biomarkers for clinical use. This review provides a comprehensive overview of experimental studies on the role of circular RNAs in ischemic stroke. Their involvement will be explored across multiple experimental models of acute ischemic stroke”.